# Regulation of the Mitochondrial BK_Ca_ Channel by the Citrus Flavonoid Naringenin as a Potential Means of Preventing Cell Damage

**DOI:** 10.3390/molecules25133010

**Published:** 2020-06-30

**Authors:** Anna Kicinska, Rafał P. Kampa, Jan Daniluk, Aleksandra Sek, Wieslawa Jarmuszkiewicz, Adam Szewczyk, Piotr Bednarczyk

**Affiliations:** 1Department of Bioenergetics, Adam Mickiewicz University, 61-614 Poznan, Poland; anias@amu.edu.pl (A.K.); wiesiaj@amu.edu.pl (W.J.); 2Department of Physics and Biophysics, Institute of Biology, Warsaw University of Life Sciences—SGGW, 02-776 Warsaw, Poland; r.kampa@nencki.edu.pl; 3Laboratory of Intracellular Ion Channels, Nencki Institute of Experimental Biology, 02-093 Warsaw, Poland; jan21.daniluk09@gmail.com (J.D.); a.sek@nencki.edu.pl (A.S.); a.szewczyk@nencki.edu.pl (A.S.); 4Faculty of Chemistry, University of Warsaw, 02-776 Warsaw, Poland

**Keywords:** endothelium, mitochondria, potassium channels, naringenin, apoptosis/necrosis

## Abstract

Naringenin, a flavanone obtained from citrus fruits and present in many traditional Chinese herbal medicines, has been shown to have various beneficial effects on cells both in vitro and in vivo. Although the antioxidant activity of naringenin has long been believed to be crucial for its effects on cells, mitochondrial pathways (including mitochondrial ion channels) are emerging as potential targets for the specific pharmacological action of naringenin in cardioprotective strategies. In the present study, we describe interactions between the mitochondrial large-conductance calcium-regulated potassium channel (mitoBK_Ca_ channel) and naringenin. Using the patch-clamp method, we showed that 10 µM naringenin activated the mitoBK_Ca_ channel present in endothelial cells. In the presence of 30 µM Ca^2+^, the increase in the mitoBK_Ca_ channel probability of opening from approximately 0.25 to 0.50 at −40 mV was observed. In addition, regulation of the mitoBK_Ca_ channel by naringenin was dependent on the concentration of calcium ions. To confirm our data, physiological studies on the mitochondria were performed. An increase in oxygen consumption and a decrease in membrane potential was observed after naringenin treatment. In addition, contributions of the mitoBK_Ca_ channel to apoptosis and necrosis were investigated. Naringenin protected cells against damage induced by tumor necrosis factor α (TNF-α) in combination with cycloheximide. In this study, we demonstrated that the flavonoid naringenin can activate the mitoBK_Ca_ channel present in the inner mitochondrial membrane of endothelial cells. Our studies describing the regulation of the mitoBK_Ca_ channel by this natural, plant-derived substance may help to elucidate flavonoid-induced cytoprotective mechanisms.

## 1. Introduction

Naringenin, a metabolite of naringin, belongs to a large group of polyphenolic compounds that are widely distributed in all food products of plant origin. Naringenin is the predominant flavanone found in grapefruit, kumquat, tomato skin, bergamot and many other fruits and herbs [1]. Naringenin has been shown to have various beneficial effects on cells both in vitro and in vivo, e.g., antiviral, anticancer, antidiabetic, anti-inflammatory and cardioprotective properties [2]. Biological activity, bioavailability and low toxicity give broad prospects for using this substance as a potential drug in many human diseases. Although the antioxidant activity of naringenin has long been thought to be a crucial factor in its cellular effects, mitochondrial pathways (including mitochondrial ion channels) are currently emerging as potential targets for the specific pharmacological action of naringenin in anti-ischemic, cardioprotective strategies [3,4,5].

Endothelial cells play an important physiological role in vascular homeostasis [6]. A variety of different potassium channels, including large-, intermediate- and small-conductance (BK_Ca_, IK_Ca_ and SK_Ca_) channels, are expressed in the plasma membrane of vascular wall cells [7]. Some studies suggest that BK_Ca_ channels are preferentially expressed in the vascular smooth muscle cell plasma membrane, while IK_Ca_ and SK_Ca_ are usually located in the endothelial cell plasma membrane [8]. It has been proposed that endothelial potassium channels are implicated in the control of vascular tone, e.g., the release of NO and endothelium-derived hyperpolarizing factor [9,10]. A large-conductance calcium-regulated potassium channel (mitoBK_Ca_ channel) was also described in the inner mitochondrial membrane of endothelial cells [11]. This is an important discovery, because the role of the mitochondria in endothelial dysfunction is now well appreciated [12].

Recently, the contribution of mitochondrial potassium channels present in the inner mitochondrial membrane to different physiological processes in the cell has been described [13]. The role of these channels in regulating the matrix volume and thus increasing fatty acid oxidation, oxidative phosphorylation and ATP synthesis [14], preventing Ca^2+^ overload [15], apoptosis, necrosis and elevated reactive oxygen species generation [16] has been proposed. Additionally, it has been shown that ischemic preconditioning, a phenomenon in which short episodes of ischemia increase tissue tolerance to subsequent lethal insult, can be mimicked by administration of potassium channel openers [17]. It is generally accepted that primary cytoprotective effects are derived from potassium channels present in the inner mitochondrial membrane [18,19].

The mitoBK_Ca_ channel is the most common type of potassium channel found in the inner mitochondrial membrane. It has been found in the inner mitochondrial membrane of the human glioma LN229 cell line [20], skeletal muscle [21], brain [22,23], heart [24,25], endothelial cells [11] and dermal fibroblasts [26]. The mitoBK_Ca_ channel is activated by NS1619 and NS11021 and blocked by paxilline, iberiotoxin or charybdotoxin [27]. The molecular components of the mitoBK_Ca_ channels are currently under debate. Recent observations suggest that these channels are structurally similar to plasma membrane BK_Ca_ channels [19,28]. The functional BK_Ca_ channel is a tetramer of the pore-forming α subunit. However, depending on the tissue, the α subunits may be associated with distinct auxiliary β subunits that determine the various physiological functions of BK_Ca_ channels [29].

Given the above findings, the current study addresses the effects of the citrus flavonoid naringenin on the mitoBK_Ca_ channel present in the mitochondria of endothelial cells. In our experimental model, we also examine in detail the functional role of the mitoBK_Ca_ channel. The presented study complements the previous biochemical and functional characteristics of the mitoBK_Ca_ channels present in fibroblast cells [30] and in vascular smooth muscle cells [31].

## 2. Results

### 2.1. Regulation of the mitoBK_Ca_ Channel by Ca^2+^ and NS11021

To obtain and describe the biophysical and pharmacological properties of the mitoBK_Ca_ channel present in the mitochondria of vascular endothelial cells, a patch-clamp ion current measurement technique was used. The process of mitoplast preparation from the mitochondria of EA.hy926 cells and the course of the experiment is schematically depicted in Figure 1a. After adding the mitochondrial suspension to hypotonic medium, mitochondrial swelling begins due to osmotic shock, which causes a rupture of the outer membrane and the formation of vesicles deprived of the outer membrane, i.e., mitoplasts. The choice of mitoplasts was based on their characteristic features: a sphere-like shape, with black dots on the surface and a characteristic "cap", being a remnant of the mitochondrial outer membrane cracks. During the experiments using the patch-clamp technique, we tested 73 mitoBK_Ca_ channels, which constituted approximately 20% of all the samples (the remaining samples were empty patches or patches containing other ion channels).

The mitoBK_Ca_ channels were identified based on the measurement of their single channel conductivity (270 ± 2 pS) and their characteristic opening patterns. The conductance of the channel was calculated from the current–voltage relationship as was shown in our previous studies [11]. Measurements were carried out with two output potentials, −40 mV and +40 mV (potential applied), and at various concentrations of calcium ions (10, 30 and 100 μM Ca^2+^). The mitoBK_Ca_ channel activity depends on the concentration of calcium ions and the voltage. To confirm the calcium sensitivity, patch-clamp control experiments were performed (Figure 1b,e). The characteristic single channel recordings at given conditions are shown in Figure 1b. Under physiological conditions (100 μM Ca^2+^), the probabilities of channel opening (Po) of 0.86 ± 0.08 at +40 mV and 0.33 ± 0.21 at −40 mV were observed (Figure 1e). Gradual lowering of the calcium ion concentration caused a reduction in the probability of opening compared to the control (100 μM Ca^2+^). At 30 μM Ca^2+^, the probabilities of opening of 0.61 ± 0.33 (+40 mV) and 0.26 ± 0.12 (−40 mV) were observed, with probabilities of 0.42 ± 0.23 (+40 mV) and 0.05 ± 0.04 (−40 mV) at 10 μM Ca^2+^ and 0.06 ± 0.08 (+40 mV) and 0.001 (−40 mV) at 1 μM Ca^2+^. A return to the control conditions of 100 μM Ca^2+^ (washout) resulted in an increase in the channel activity (Po, 0.87 ± 0.08 (+40 mV) and 0.26 ± 0.12 (−40 mV)) (Figure 1e). The effects of 1 µM NS11021 as a characteristic activator of mitoBK_Ca_ channels and 1 μM of paxilline, which is a potent inhibitor of the channel, were also examined (Figure 1c,d). Single channel recordings are shown in Figure 1c. Under control conditions, the Po of the channel was 0.29 ± 0.11 at −40 mV. After the application of NS11021, the probability of opening increased to 0.64 ± 0.10, while paxilline inhibited the channel (Po = 0.004 ± 0.004) (Figure 1d).

The effect of 1, 3 and 10 μM naringenin on the activity of the single mitoBK_Ca_ channel was then examined (Figure 2). Representative single channel recordings are shown in Figure 2a (conditions as indicated). The addition of naringenin, at –40 mV and in the presence of 10 μM Ca^2+^, increased the probability of opening of the mitoBK_Ca_ channel in a dose-dependent manner from a control value of 0.05 ± 0.04 up to 0.23 ± 0.08 (10 μM naringenin). After washout, the Po returned to 0.04 ± 0.04 (Figure 2b, left panel). At +40 mV, the Po increased from a control value of 0.41 ± 0.17 up to 0.68 ± 0.05 with increasing naringenin concentrations, returning to 0.45 ± 0.14 after washout (Figure 2b, left panel). The dose-dependent increase in the Po of the channel was also observed after naringenin addition in the presence of 30 μM Ca^2+^; from 0.26 ± 0.13 to 0.47 ± 0.06 and 0.61 ± 0.29 to 0.75 ± 0.01 at −40 and +40 mV, respectively (Figure 2b, middle panel). In the presence of 100 μM Ca^2+^, 3 µM naringenin increased the Po at −40 mV from 0.33 ± 0.11 to 0.61 ± 0.05. No further stimulation was observed with 10 µM naringenin (Figure 2b, right panel). At +40 mV, naringenin did not change the activity of the mitoBK_Ca_ channel. Taken together, significant effects of mitoBK_Ca_ channel activation were observed in the presence low calcium concentration.

### 2.2. Effect of Naringenin on Respiratory Rate and ∆Ψ of Isolated Endothelial Mitochondria

To investigate the effect of naringenin on potassium permeability in endothelial mitochondria, the nonphosphorylating (resting) oxygen consumption rate and membrane potential (ΔΨ) of mitochondria isolated from EA.hy926 cells were measured in potassium-containing media. As shown in Figure 3a, the addition of 5–50 μM naringenin significantly and dose-dependently decreased the mitochondrial ΔΨ by 0.65 ± 0.58 up to 7.01 ± 0.84 mV. The stimulatory effect of naringenin was partially reversed by the mitoBK_Ca_ potassium channel inhibitors iberiotoxin (IbTx) (2 µM) and paxilline (0.3 mM). The addition of inhibitors decreased the effect of naringenin up to ∼85% and ∼46%, respectively. As shown in Figure 3b, the addition of 5–50 μM naringenin also increased the respiratory rate, depending on the dose, by 1.47 ± 4.38 up to 7.27 ± 4.88 nmol O_2_ × min^−1^ × mg protein^−1^. The stimulatory effect of naringenin was partially reversed by 2 µM IbTx and 0.3 mM paxilline. However, these effects were not statistically significant (*p* > 0.05). Because a saturation of the effect was observed at 10 μM naringenin, this concentration was used in further experiments with isolated mitochondria. The results from Figure 3 suggest that naringenin stimulates inhibitor-sensitive mitoBK_Ca_ channel-mediated K^+^ flux into endothelial mitochondria, decreasing the ∆Ψ and thus accelerating the mitochondrial respiration rate.

The modulation of the naringenin effect on endothelial mitochondria by Ca^2+^ is shown in Figure 4. Because residual Ca^2+^ is present under experimental conditions (derived from mitochondria), the addition of 1.5 mM ethylene glycol tetraacetic acid (EGTA) (a chelator of Ca^2+^) decreased the nonphosphorylating respiratory rate slightly by ~4% (to 23.5 nmol O_2_ × min^−1^ × mg protein^−1^) compared to the control (24.4 nmol O_2_ × min^−1^ × mg protein^−1^) (Figure 4a). In the presence of 100 μM Ca^2+^, the nonphosphorylating respiratory rate was increased by ∼8% (to 26.4 nmol O_2_ × min^−1^ × mg protein^−1^), as expected. Naringenin (10 µM) significantly increased the respiratory rate in mitochondria by ∼15.6% (to 28.2 nmol O_2_ × min^−1^ × mg protein^−1^) and ∼12.5% (to 26.4 nmol O_2_ × min^−1^ × mg protein^−1^), respectively, at low concentrations of Ca^2+^ (no additions) and in the presence of 1.5 mM EGTA. In the presence of 2 µM IbTx, naringenin-induced stimulation of respiration was diminished in both conditions by ∼50% (by 53% at low Ca^2+^ and 46% with EGTA), again suggesting the involvement of the mitoBK_Ca_ channel in the observed phenomena. The effect of naringenin on the nonphosphorylating respiratory rate was very small in the presence of 100 μM Ca^2+^. As shown in Figure 4b, 10 µM naringenin induced a slight decrease in the ∆Ψ under all experimental conditions. The effect was most visible under low Ca^2+^ conditions (no additions) (1.12 ± 0.5 mV) and slightly less apparent in the presence of 1.5 mM EGTA (0.74 ± 0.4 mV) or 100 μM Ca^2+^ (0.85 ± 0.3 mV). The naringenin-induced depolarization of the inner mitochondrial membrane was diminished in the presence of 2 µM IbTx (by 63, 23 and 61% under low Ca^2+^, 1.5 mM EGTA, and 100 µM Ca^2+^ conditions, respectively), although only the results obtained under low Ca^2+^ conditions were statistically significant.

### 2.3. Cytotoxicity Tests and Cytoprotective Effects of Naringenin

Apoptosis and necrosis tests were performed to check the potential toxicity of naringenin. Cells were treated with naringenin and a quantitative assessment of apoptosis and necrosis was performed at given time points (Figure 5a). At both concentrations of naringenin (10 and 30 μM), no induction of apoptosis or necrosis was observed, therefore, no naringenin toxicity was detected (Figure 5a, c). Naringenin has previously been shown to have cytoprotective properties [32]. A one-hour preincubation with naringenin prior to the administration of tumor necrosis factor α (TNF-α) together with cycloheximide (CHX) (factors that induce endothelial cell apoptosis) [33] resulted in a significant dose-dependent decrease in apoptosis (by ~25% and ~40% for 10 and 30 μM naringenin, respectively). Under these conditions, no necrotic cell death was observed up to 13 h (Figure 5a, right panel). However, after 24 h, necrotic cell death occurred in ~35% of TNF-α/CHX-treated cells and a slight protection by 30 μM naringenin was apparent (Figure 5a, right panel, inset). Representative microscopic images showing the effect of naringenin on EA.hy926 cells treated with TNF-α/CHX or untreated are shown in Figure 5c. The properties of naringenin were compared with the potent mitoBK_Ca_ channel activator, NS11021 (Figure 5b). NS11021 (3 µM) did not cause apoptotic or necrotic cell death and provided ~50% protection against TNF-α/CHX-induced apoptosis (Figure 5b). The effect of NS11021 on TNF-α/CHX-induced necrosis is also shown in the inset of Figure 5b, right panel. NS11021 reduced necrosis by ~15%.

## 3. Discussion

Increasing evidence supports the involvement of potassium channels present in the inner mitochondrial membrane in cellular life/death cycles [34]. For example, the role of mitochondrial potassium channels in ischemia/reperfusion-related cytoprotection in the mammalian myocardium [18,35,36] and in brain tissue [37,38] has been reported. Activation of mitochondrial potassium channels by potassium channel openers in tissues before or during the onset of ischemia induces cytoprotection. Several hypotheses have been proposed regarding the mechanism of protection through the activation of mitochondrial potassium channels, but these studies are yet not conclusive [39,40].

The mitoBK_Ca_ was found in the inner mitochondrial membrane of the different cells type. It is well accepted that the function, localization and regulation of BK_Ca_ channels depend on the expression of specific alternative splice variants of the pore-forming α subunit (single gene) and the associated β regulatory subunits [29]. Recently, mitoBK_Ca_ channels in cardiomyocytes, which are encoded by the KCNMA1 gene that encodes the plasma membrane BK_Ca_ channels, have been shown to be implicated as infarct-limiting factors in ischemia/reperfusion injury [19,28].

It has been shown that mitoBK_Ca_ channels (like the plasma membrane BK_Ca_ channels) are sensitive to potassium channel openers (NS1619 or NS11021) and potassium channel blockers (paxilline and iberiotoxin) [18,41]. However, the majority of mitochondrial potassium channel modulators exhibit a broad spectrum of off-target effects [42,43]. These include uncoupling properties, respiratory chain inhibition and effects on calcium homeostasis in cells [13,44,45]. Therefore, there is a great need to find new agents specific for mitochondrial potassium channels, especially agents of natural origin [46]. Furthermore, the rational use of specific channel inhibitors or activators is critical to understanding the consequences of mitochondrial channel inhibition or activation in cells.

The broad spectrum of biologically active flavonoids widely distributed in the plant kingdom, including grapefruit (*Citrus paradisi*) compounds such as naringenin, have previously encouraged researchers to study their various beneficial pharmacological effects, i.e. anti-inflammatory, anticancer or antioxidative properties. Recently, flavonoids have attracted attention as new potassium channel modulators. For example, naringenin, a flavonoid used in many traditional Chinese herbal medicines, has been shown to relax various smooth muscles [47]. Furthermore, it has been proposed that in mice, naringenin reduces superoxide anion-induced inflammatory pain by activating the plasma membrane K_ATP_ channel [48]. Other studies have shown that naringenin-induced activation of the plasma membrane potassium channel is likely to affect neuronal function [49]. Naringenin has also been reported to inhibit cardiac human *ether-a-go-go*-related gene (HERG) channels similar to well-characterized pharmaceutical HERG antagonists [50]. In addition, the additive inhibitory effect on HERG currents was investigated by combining naringenin with antiarrhythmic drugs [51]. This additive HERG inhibition may cause an increased risk of arrhythmia by increasing repolarization delay and possible repolarization heterogeneity.

In mitochondria, flavonoids have been proposed to have multiple targets including oxidative phosphorylation [52], apoptotic pathways [53] and mitochondrial ion channels [4,54]. However, the effects of flavonoids on mitochondrial potassium channel activity has only been indirectly reported using biochemical techniques.

The presence of the mitoBK_Ca_ channel in the mitochondria of human endothelial cells is very well-documented [11,55]. In our current studies, the application of the patch-clamp technique and single channel recordings of mitochondrial potassium channels was a key approach to identify the interaction of naringenin with the mitoBK_Ca_ channel. Using this methodology, we have obtained direct evidence that naringenin has properties that activate the mitoBK_Ca_ channel present in the mitochondria of human endothelial cells. The stimulatory effect of naringenin on the mitoBK_Ca_ channel was also confirmed by measurements of membrane depolarization and changes in respiratory rate in endothelial mitochondria. Naringenin stimulated mitochondrial respiration and decreased ΔΨ. The effects of naringenin were diminished by the use of mitoBK_Ca_ channel inhibitors such as iberiotoxin and paxilline. It has previously been shown that activation of the channel followed by an influx of K^+^ into the mitochondrial matrix results in an increase in the mitochondrial respiration measured in cells [56]. Thus, electrophysiological and biochemical studies strongly confirm the stimulatory action of naringenin on the mitoBK_Ca_ channel present in endothelial cells.

In our previous study, we observed a similar interaction of naringenin with the mitoBK_Ca_ channel identified in the mitochondria of human skin fibroblasts [30]. However, the stimulatory effects of naringenin on the mitoBK_Ca_ channel in skin fibroblasts were stronger than those observed in endothelial cells. The differences in mitoBK_Ca_ channel regulation by naringenin are most likely due to the presence of tissue-specific splice variants that encode the channel protein and/or the expression of different β regulatory subunits. In the mitochondria of skin fibroblasts, β3 regulatory subunits were identified, predominantly [26]. On the other hand, presence of the auxiliary β2-subunit of the channel in the endothelial mitochondrial inner membrane was confirmed [11]. Clarification of these differences in the regulation of the mitoBK_Ca_ channel by naringenin requires further studies.

The cytoprotective effects of naringenin have been observed in various cell models [57]. In the present study, a human endothelial cell line (EA.hy926 cells) and TNF-α/CHX-induced stress [33,58] were used to explore the protective effects of naringenin. In our experimental model of TNF-α/CHX-induced apoptosis/necrosis, naringenin exhibited significant cytoprotective effects. Similar but more pronounced effects were observed when NS11021 was used as a mitoBK_Ca_ channel activator. The protective effect of mitoBK_Ca_ channel stimulation by naringenin against ischemia/reperfusion injury has also been suggested in Langendorff-perfused hearts [4]. In addition, other studies have shown the vasorelaxant effect of naringenin on endothelium-denuded vessels, which was due to the activation of BK_Ca_ channels in myocytes [31]. It has also been previously shown that the adhesion of monocytes to endothelial cells, an early proinflammatory step in artherogenesis, is reduced by treatment with naringenin [59,60]. Although, our study indicates that the activation of the mitoBK_Ca_ channel is a relevant mechanism of action that can significantly contribute to the cardioprotective effects of naringenin at the cardiovascular level, the involvement of other mechanisms such as antioxidant, vasorelaxant and antiatherosclerotic cannot be excluded.

The protective effects of naringenin observed previously and in our current data confirm that naringenin can penetrate the membranes of endothelial cells, including the plasma and inner mitochondrial membranes. Naringenin accumulation in other cells has also been observed [5,61]. In addition, flavonoids have been shown to cross the blood–brain barrier and into various brain regions, and efflux transporters have been described to limit their entry [62]. Naringenin has high permeability in in vitro and in situ blood–brain barrier models, suggesting that flavonoids from this family may also have protective effects on the nervous and cardiovascular systems.

## 4. Materials and Methods

### 4.1. Cell Culture

The stable human endothelial cell line EA.hy926 (Cat. No. ATCC®CRL-2922™), originally derived from a human umbilical vein, was used [11,63]. Cells were grown in Dulbecco’s modified Eagle’s medium (1000 mg/L D-glucose) supplemented with 10% fetal bovine serum (FBS), 1% L-glutamine, 2% hypoxanthine-aminopterin-thymidine (HAT), 1% penicillin/streptomycin in a humidified 5% CO_2_ atmosphere, at 37 °C. For bioenergetic measurements, the EA.hy926 cells were cultured for six days in 150 mm dishes until they reached approximately 90–100% confluence. For electrophysiology experiments, the cells were reseeded every third day. Cells between passages 5 and 20 were used in this study.

### 4.2. Mitochondrial Preparation for Bioenergetics Measurements

Mitochondria were isolated as previously described, with minor modifications [64]. For each experiment, the cell cultures from 100 dishes (150 mm) were harvested with trypsin/EDTA, rinsed twice with phosphate-buffered saline (PBS; 10% and 5% FBS in the first and second wash, respectively), and centrifuged at 1500× *g* for 10 min. The cells were resuspended in PREPI medium (0.25 M sucrose, 1.5 mM EDTA, 1.5 mM EGTA, 0.2% BSA and 15 mM Tris–HCl, pH 7.2) at a ratio of 3 ml of medium per 1 g of cells. The cells were then homogenized by ten passes with a tight Dounce homogenizer (GPE Scientific, Leighton Buzzard, UK), and the homogenate was subsequently centrifuged at 1200× *g* for 10 min. The cell pellets were resuspended and were again homogenized (ten passes) and centrifuged at 1200× *g* for 10 min. The pellets were resuspended, and the cells were once again homogenized (eight passes) and centrifuged at 1200× *g* for 10 min. The supernatants from the three homogenization steps were centrifuged at 12,000× *g* for 10 min. The mitochondrial pellets were washed with a PREPII medium containing 0.25 M sucrose and 15 mM Tris–HCl, pH 7.2, and centrifuged at 700× *g* for 7 min. The resulting supernatant was centrifuged at 12,000× *g* for 10 min. The final mitochondrial pellet was resuspended in a small volume of PREPII medium [64]. All of the steps were performed at 4 °C. Protein concentration was determined using the Bradford method (Bio-Rad, Hercules, California, US). The yields of isolated mitochondria were equal to 4.7 ± 0.9 mg mitochondrial protein/g cells (*n*  =  12).

### 4.3. Mitochondria and Mitoplast Preparation for Electrophysiology

For electrophysiological measurements, fresh mitochondria and subsequent mitoplasts were prepared by differential centrifugation and hypotonic swelling, respectively, as previously described [11,65]. Mitoplasts were prepared from the EA.hy926 mitochondria by incubation in a hypotonic solution (5 mM HEPES, 100 µM CaCl_2_, pH 7.2) for approximately 1 min, and then a hypertonic solution (750 mM KCl, 30 mM HEPES, and 100 µM CaCl_2_, pH 7.2) was subsequently added to restore the isotonicity of the medium (*n* = 90). For each/repeating patch-clamp experiment, a fresh mitoplast preparation was used.

### 4.4. Patch-Clamp Experiments

Patch-clamp experiments using endothelial mitoplasts were performed as previously described [11,65]. In brief, a patch-clamp pipette was filled with an isotonic solution containing 150 mM KCl, 10 mM HEPES, and 100 µM CaCl_2_ at pH 7.2. All of the modulators of the mitoBK_Ca_ channel were added as dilutions in the isotonic solution. To apply these substances, a perfusion system was used. The mitoplasts at the tip of the measuring pipette were transferred into the openings of a multibarrel “sewer pipe” system in which their outer faces were rinsed with the test solutions (Figure 1a). The current–time traces of the experiments were recorded in a single-channel mode. The pipettes were made of borosilicate glass, had a resistance of 10–20 MΩ (Harvard Apparatus GC150-10, Holliston, Massachusetts, USA). The currents were low-pass filtered at 1 kHz and sampled at a frequency of 100 kHz (amplifiers: Axopatch™ 200B, digidata: Axon 1440A—Axon Instruments, Molecular Device, San Jose, California, USA). The traces of the experiments were recorded in single-channel mode. The illustrated channel recordings are representative of the most frequently observed conductance for a given condition. The conductance of the channel was calculated from the current-voltage relationship [11]. The open probability (Po) of the channels was determined using the single-channel search mode of the Clampfit 10.7 software (Axon Instruments, Molecular Device) [11,65].

### 4.5. Measurements of Mitochondrial Oxygen Consumption and ∆Ψ with Isolated Mitochondria

Mitochondrial respiration and ΔΨ were measured in isolated endothelial mitochondria as previously described [11,64]. Oxygen uptake was determined polarographically using a Rank Bros. (Cambridge, UK) oxygen electrode in 2.8 ml of incubation medium (50 mM KCl, 70 mM sucrose, 2.5 mM KH_2_PO_4_, 2 mM MgCl_2_, 10 mM HEPES, 10 mM Tris/HCl (pH 7.2), and 0.05% BSA) with 2 mg of mitochondrial protein (at 37 °C). Mitochondrial ∆Ψ was measured simultaneously with oxygen uptake using a tetraphenylphosphonium (TPP^+^)-specific electrode. Measurements were made in the presence of 5 mM succinate (as respiratory substrate), 4 µM rotenone (to inhibit complex I), 2 µg oligomycin (to inhibit ATP synthase), 1.7 µM carboxyatractyloside (to inhibit ATP/ADP antiporter), and 0.15 mM ATP (to activate succinate dehydrogenase). Up to 2 µM iberiotoxin (dissolved in water; Alomone Labs, Israel) or 0.3 mM paxilline (dissolved in methanol; Merck, Poznan, Poland) were used to inhibit the mitoBK_Ca_ channel.

### 4.6. Apoptosis/Necrosis Assay

Apoptosis and necrosis were detected using the RealTime-Glo™ Annexin V Apoptosis and Necrosis Assay (Promega, Walldorf, Germany). EA.hy926 cells were plated in a black 96-well assay plate (Corning Incorporated Costar®, Merck, Poznan, Poland) at a concentration of 8500 cells per well. The test was carried out after 48 h of culture in complete cell culture medium, according to the manufacturer’s instruction. Measurements were performed using the Infinite M200 pro plate reader (Tecan Group Ltd, Männedorf, Switzerland) at 0, 1, 3, 7, 9, 11, 13, and 24 h. Naringenin dilutions were prepared from a stock solution in DMSO, therefore, an additional control of 0.1% DMSO was introduced.

### 4.7. Statistical Analysis

The results are presented as the means ± SD obtained from at least three independent mitochondrial isolations (biochemical data), in which each determination was performed at least in duplicate. Electrophysiological results are shown as the means ± SD obtained from three to six independent experiments, with each determination performed at least in triplicate. An unpaired two-tailed Student’s t-test was used to identify any significant differences. For patch-clamp experiments, one-way ANOVA was used to compare the means of three or more treatment conditions. For all tests, a P value was considered to be significant if *p* < 0.05 (*), *p* < 0.01 (**), or *p* < 0.001 (***).

## 5. Conclusions

The presented data strongly suggests that the activation of the mitoBK_Ca_ channel is an important mechanism accounting for the cardioprotective effects of naringenin. These results broaden our knowledge of the regulation of the mitoBK_Ca_ channel by flavonoids and may shed light on the physiological targets of naringenin. The molecular mechanisms responsible for many of these physiological effects still need to be identified, but mitoBK_Ca_ channels should be included in the growing list of potential flavonoid targets.

## Figures and Tables

**Figure 1 molecules-25-03010-f001:**
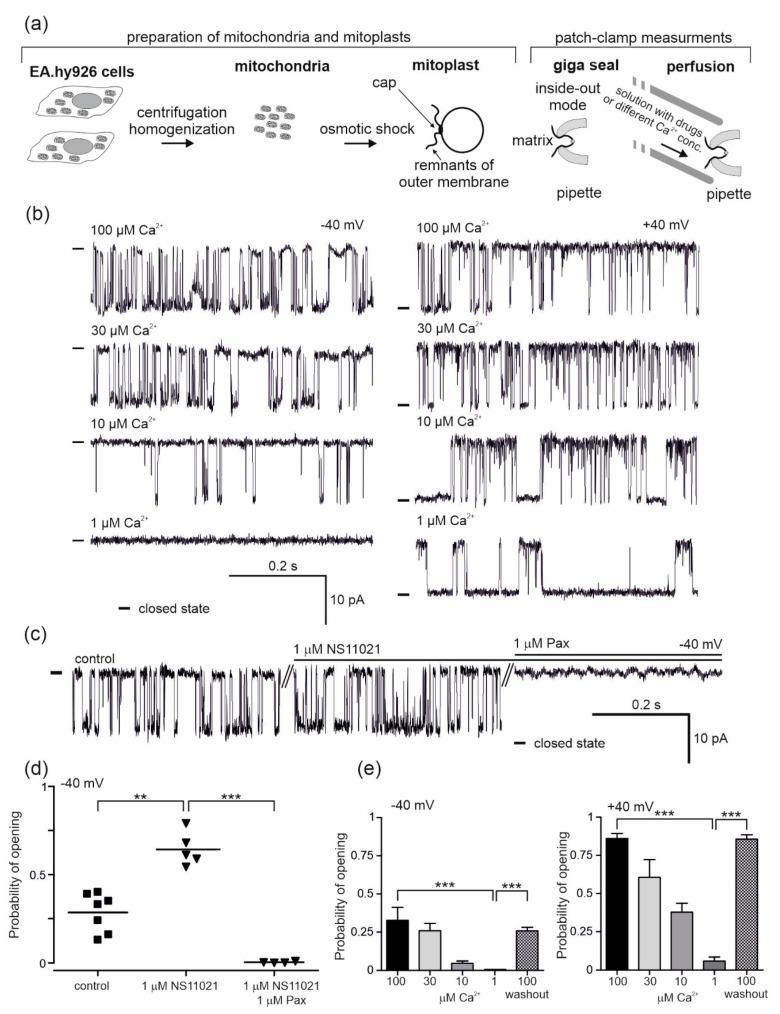
Pharmacology of the large-conductance calcium-regulated potassium channel (mitoBK_Ca_ ) from the mitochondria of human endothelial cells. (**a**) Schematic representation of mitochondria and mitoplast preparation from human endothelial cells, including mitoplast patching and the patch-clamp experiment in inside-out mode with the perfusion system. The matrix side of the mitochondrial membrane is exposed to externally added substances from the perfusion system. (**b**) Regulation of the mitoBK_Ca_ by calcium ions. Representative single-channel current-time recordings recorded at different “matrix” Ca^2+^ concentrations (1, 10, 30 and 100 μM) and applied voltages (−40 and +40 mV) (*n* = 6) are shown. “-“ indicates a closed state of the channel. (**c**) The effect of mitoBK_Ca_ modulators, NS11021 (1 µM) and paxilline (Pax; 1 µM), on the single-channel activity at −40 mV. “-“ indicates a closed state of the channel. (**d**) The influence of NS11021 (*n* = 5) and paxilline (*n* = 4) on the probability of opening of the mitoBK_Ca_ channel. (**e**) Dependence of the probability of opening of the mitoBK_Ca_ channel on Ca^2+^ concentration (*n* = 6). *** *p* < 0.001, ***p* < 0.01. All data are presented as the means ± SD.

**Figure 2 molecules-25-03010-f002:**
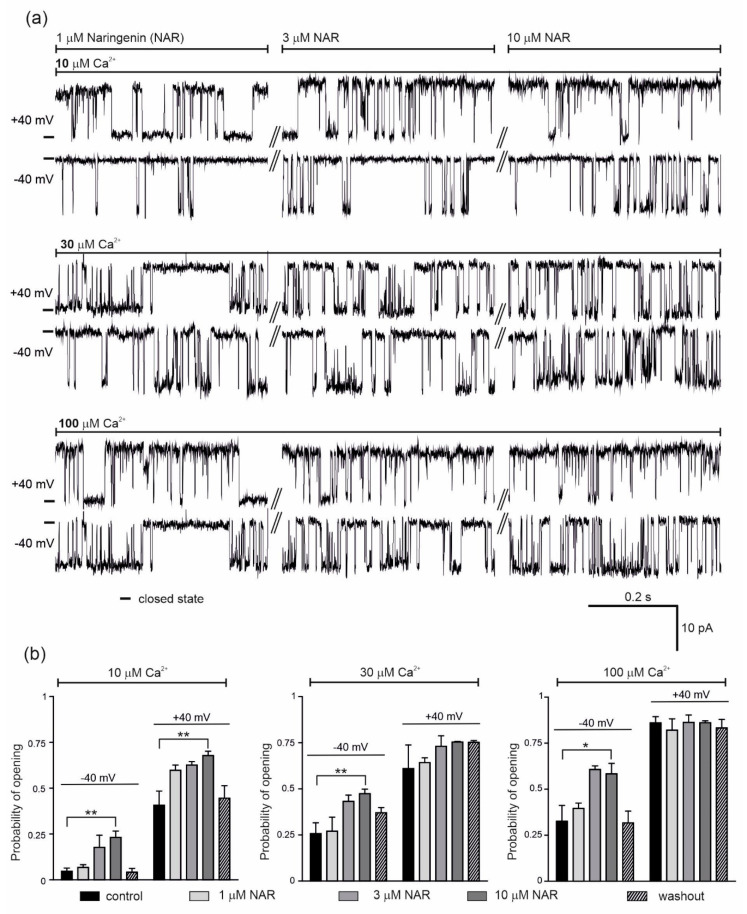
Effect of naringenin on the endothelial mitoBK_Ca_ channel activity. (**a**) Representative single-channel recordings of the mitoBK_Ca_ at −40 and +40 mV. Experiments were performed in a symmetric 150/150 KCl with additions as indicated. (**b**) The mitoBK_Ca_ probability of opening at –40 and + 40 mV in the presence of naringenin (1, 3 and 10 μM) at different Ca^2+^ concentrations, as indicated. The data are presented as the means ± SD (*n* = 5). ** *p* < 0.01, * *p* < 0.05.

**Figure 3 molecules-25-03010-f003:**
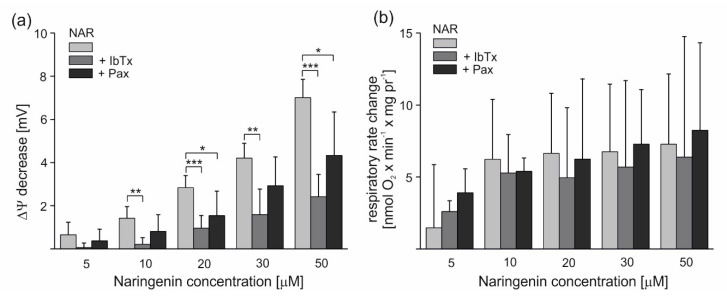
Naringenin-induced changes in the ∆Ψ and respiratory rate of isolated endothelial mitochondria under nonphosphorylating conditions. The effect of different naringenin concentrations on ∆Ψ (**a**) and respiratory rate change (**b**) in the absence or presence of mitoBK_Ca_ channel inhibitors (2 µM iberiotoxin (+IbTx) or 0.3 mM paxilline (+Pax)). The data represent at least five different mitochondrial preparations and are expressed as the means ± SD. *** *p* < 0.001, ** *p* < 0.01, * *p* < 0.05.

**Figure 4 molecules-25-03010-f004:**
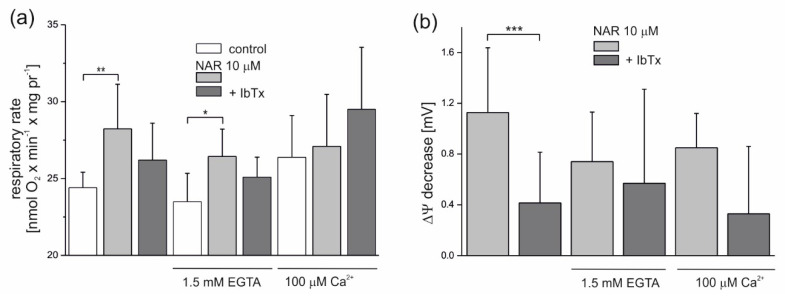
Influence of naringenin on the oxygen consumption rate and ∆Ψ of isolated endothelial mitochondria under nonphosphorylating conditions. Effect of 10 µM naringenin on mitochondrial oxygen consumption rate (**a**) and ΔΨ (**b**) under control conditions (control) and in the absence or presence of 2 µM iberiotoxin (+IbTx). The Ca^2+^ concentration was modulated by the addition of 1.5 mM ethylene glycol tetraacetic acid (EGTA) or 100 µM Ca^2+^ to the incubation medium, as indicated. The data represent at least five different mitochondrial preparations and are expressed as the means ± SD. *** *p* < 0.001, ** *p* < 0.01, * *p* < 0.05. NAR—naringenin, IbTx—iberiotoxin.

**Figure 5 molecules-25-03010-f005:**
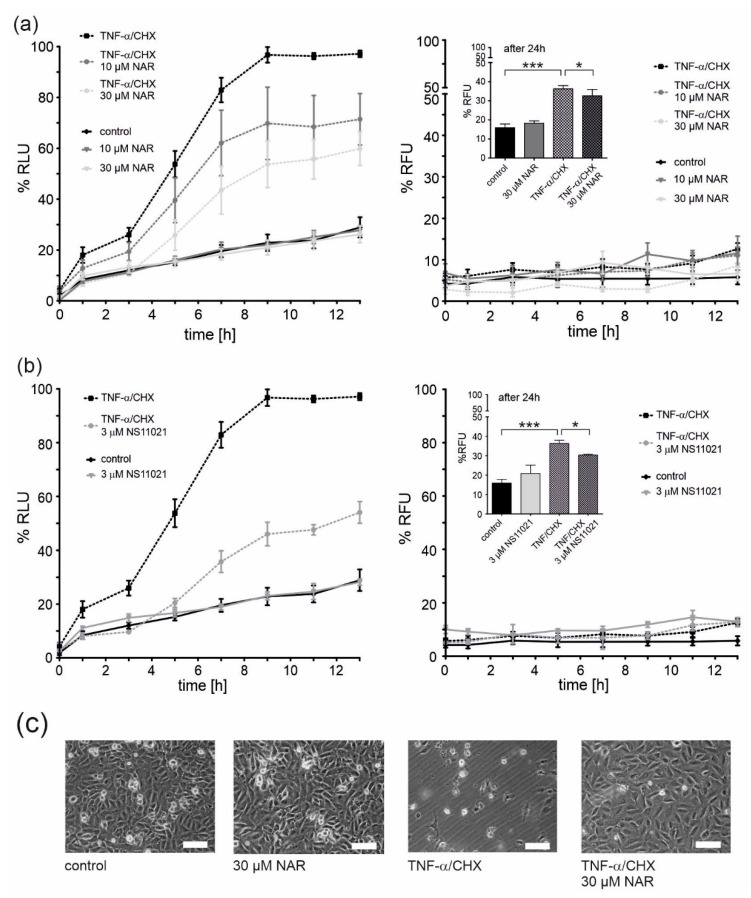
Protective effects of mitoBK_Ca_ channel activation on the viability of TNF-α/CHX-treated human endothelial cells. Cell viability was measured using the RealTime-GloTM Annexin V Apoptosis and Necrosis Assay. Cell apoptosis (left panels) and necrosis (right panels) was measured at the given time points. The effect of 10 and 30 μM naringenin (NAR) (**a**) and a mitoBK_Ca_ channel opener NS11021 (**b**) on EA.hy926 cells treated with 1 ng/ml TNF-α combined with 0.05 μg/ml cycloheximide (CHX). Data are presented as the means ± SD (*n* = 6). (**c**) Representative images of human endothelial cells. Cells were treated for 24 h with naringenin (30 μM), TNF-α (1 ng/ml)/CHX (0.05 μg/ml) or TNF-α/CHX and naringenin. Images were prepared using an inverted microscope (Olympus IX71) and DLT-Cam. Scale bar is equal to 200 μm. *** *p* < 0.001, * *p* < 0.05.

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
