# Peer review of "Regulation of the Mitochondrial BKCa Channel by the Citrus Flavonoid Naringenin as a Potential Means of Preventing Cell Damage"

_molecules, 2020, doi:10.3390/molecules25133010_

Round 1

Reviewer 1 Report

The manuscript from Kicinska et al. is an interesting work dealing with the study of the effect of the citrus flavonoid naringenin on the endothelial mitoBKCa channel and its cytoprotective effects on endothelial mitochondria.

The manuscript is properly written, techniques used are suited for the study, experiments follow a logical order, and the conclusions are mostly in agreement with the results obtained. Although the study is quite similar to a previous one from the same group dealing with the effect of naringenin on dermal fibroblasts, it is still interesting and suitable for publication, since it broadens the knowledge on the effects of this flavonoid on different tissues and thus its specificity.

However, I have some concerns about this work that authors should address before:

  • The statistical analysis is incomplete. It has been done for certain experiments but not for others. Authors should include the statistical analysis in every experiment (figure 1 e, f; figure 2b; figure 3b; inset figure 5a and b, right panel). Of course, the consequences of the statistical analysis on the results must be added to the text.
  • A dose-response graphic with the effect of naringenin of the different parameters measured would be interesting (on the open probability; depolarization and respiratory rate change). An IC50 could also be calculated in some cases and compared to other previous works with this or other similar flavonoids.
  • In some experiments the additions of the inhibitors IbTx and Pax have an effect opposite to the expected one. For example in figure 3b with 5 μM NAR, or the figure 4a with 100 μM Calcium. Any comment?
  • The effect on the respiratory rate change of NAR seem to saturate at 10 μM which is not the case for depolarization. This should be discussed.
  • In figure 2b, the colour of the bar corresponding to 10μM NAR is not the same as that in the legend at the bottom of the graph.
  • In the legend of the figures 3a and b, NAR is not aligned with its symbol.
  • In figure 4a, the control with no EGTA or Calcium added has not error bar. In addition, in the legend of this figure (4a and b) NAR 10 μM is not aligned with its symbol.
  • In the legend of figure 5 it is said that the right panel corresponds to apoptosis and the left one to necrosis but it is the opposite.
  • Which is the calcium concentration in the cytotoxicity tests (figure 5)?
  • In the section 4.4, authors should add some more information about the sampling rate and the filtering of the data.

Author Response

Dear Reviewer,

The authors have carefully considered the stimulating criticism of the Referees. We have changed the manuscript at the points mentioned. Our revision includes all changes. We are convinced that the Reviewers’ suggestions have improved the quality of this manuscript.

Response to reviewers’ comments

(Reviewer 1)

  • The statistical analysis is incomplete. It has been done for certain experiments but not for others. Authors should include the statistical analysis in every experiment (figure 1 e, f; figure 2b; figure 3b; inset figure 5a and b, right panel). Of course, the consequences of the statistical analysis on the results must be added to the text.

-In the figure 1d, 1e figure 2b statistical analysis were performed and marked.

-The effects shown in figure 3b are not statistically significant (P> 0.05). The appropriate comment was added into the text: line 157 “However, these effects were not statistically significant (P > 0.05).”

-In the figure 5a statistical analysis were performed and marked as significant effects.

  • A dose-response graphic with the effect of naringenin of the different parameters measured would be interesting (on the open probability; depolarization and respiratory rate change). An IC50 could also be calculated in some cases and compared to other previous works with this or other similar flavonoids.

We agree with the Reviewer that a dose-response analysis for naringenin would be interesting. However, in the case of the open probability, there are only three concentrations of 1, 3, 5 uM, and therefore IC50 determination is impossible. When measuring mitochondrial oxygen consumption and changes in membrane potential, five concentrations (5, 10, 20, 30, 50 uM) were used. But from the concentration of 20 µM (as we wrote below) we observed a non-specific non-ionophoric effect of flavonide (with saturation of oxygen consumption there was a further decrease in membrane potential). Therefore, the IC50 calculation for 2 points (5, 10 uM) is also impossible.

  • In some experiments the additions of the inhibitors IbTx and Pax have an effect opposite to the expected one. For example in figure 3b with 5 μM NAR, or the figure 4a with 100 μM Calcium. Any comment?

The mentioned effects are not statistically significant (P > 0.05).

  • The effect on the respiratory rate change of NAR seem to saturate at 10 μM which is not the case for depolarization. This should be discussed.

Indeed, the stimulation of respiration by naringenin saturates at 10 µM, which is not the case for depolarization. This implies that the naringenin effect, at concentrations higher than 10 µM, is not a pure ionophoric effect. Thus, in the following experiments described, the concentration of 10 µM was not exceeded.

  • In figure 2b, the color of the bar corresponding to 10μM NAR is not the same as that in the legend at the bottom of the graph.

The color of the bar was corrected.

  • In the legend of the figures 3a and b, NAR is not aligned with its symbol.
  • In figure 4a, the control with no EGTA or Calcium added has not error bar. In addition, in the legend of this figure (4a and b) NAR 10 μM is not aligned with its symbol.

The figures have been adjusted as suggested by the Reviewer. The error bar in figure 4a has been corrected.

  • In the legend of figure 5 it is said that the right panel corresponds to apoptosis and the left one to necrosis but it is the opposite.

It was replaced.

  • Which is the calcium concentration in the cytotoxicity tests (figure 5)?

The RealTime-Glo ™ Annexin V Apoptosis and Necrosis Assay (by Promega) used to determine apoptosis and necrosis does not determine the exact concentration of calcium ions in the test. However, the manufacturer states that the initial CaCl2 stock is concentrated 1000x, and the final concentration of this compound in a single well of a 96-well plate is diluted to 1x. In addition, calcium ions are only needed to create the conditions for the binding of Annexin V subunits, which is a key element in detecting apoptosis in the assay. The whole test is dissolved in a standard DMEM cell culture medium, the same as is used for cell culture for all experiments of this work.

  • In the section 4.4, authors should add some more information about the sampling rate and the filtering of the data.

This issue was added: line starting 354.

Reviewer 2 Report

Kicinska, Kampa and collaborators examined the effects of a flavanone naringenin on mitochondrial BKCa (mitoBKCa) channels present in the inner mitochondrial membrane of human vascular endothelial cells by using the inside-out patch-clamp technique. First, the authors demonstrated that mitoBKCa channel currents depend on membrane potentials and Ca2+ concentrations; this channel is activated by a mitoBKCa channel activator NS11021 and is inhibited by a mitoBKCa channel inhibitor paxilline (Fig. 1). Then, they found out that naringenin dose-dependently activates mitoBKCa channels in a manner dependent on membrane potentials and Ca2+ concentrations (Fig. 2). Moreover, naringenin was found to dose-dependently depolarize mitochondrial membranes in a manner sensitive to paxilline and another mitoBKCa channel inhibitor iberiotoxin (IbTx) and to increase mitochondrial respiratory rate with a small dose-dependency (Fig. 3). The mitochondrial membrane depolarization and respiratory rate increase produced by naringenin was affected by Ca2+, albeit its effect was small (Fig. 4). The authors also revealed that naringenin dose-dependently inhibits an apoptosis induce by TNFa and cycloheximide and that a similar inhibitory effect is produced by NS11021 (Fig. 5). As a result, it was suggested that mitoBKCa channel activation produced by naringenin is a molecular mechanism for its cardioprotective effect. A similar action of naringenin on mitoBKCa channel has been reported by the authors themselves in skin fibroblasts. There are several concerns that should be addressed and are several points that may serve to amend this manuscript, as follows:

Major comments:

  1. Line 111: there is no explanation about how 270 ± 2 pS was obtained. Please amend this point.
  2. The last paragraph on page 4 and Fig. 2(b): statistical significance about dose-dependency for the naringenin effect is not clear. Please make this point clear.
  3. Line 147: ΔΨ should be defined here but not in line 353. Membrane but not membrane potential is depolarized. Line 151 and ΔΨ depolarization in the Y-axis of Fig. 3(a) should be revised.
  4. Line 153: it is not clear why paxilline at 0.3 mM was used. In line 124, paxilline (1 mM) is used. Alternatively, IbTx is used at 2 mM, a much larger value than that (0.3 mM) of paxilline. Please make these points clear.
  5. Line 157 and 158: it is not clear from Fig. 3(b) whether a saturation effect is observed at 10 mM naringenin (the effect of naringenin at 10 mM is statistically larger in extent than that at 5 mM). Please make this point clear statistically.
  6. The most left hand-side column of Fig. 4(a) does not appear to have SD. Please check whether this is right. The values of SDs of the results shown in Fig. 4 are very large. The last paragraph on page 6 should be revised by considering such large SD values.
  7. Content in lines 231-234 in Discussion section is quite the same as that in lines 70-72 in Introduction section. This point should be corrected.
  8. Lines 280-282: it would be better to mention a detail about the difference between vascular endothelial cells and skin fibroblasts (authors’ previous study) in mitoBKCa channel regulation by naringenin.
  9. Do flavonoids other than narigenin activate mitoBKCa channels? Which structure of narigenin is involved in the activation of mitoBKCa channels? Please discuss this point.
  10. Line 359: please mention the reason why rotenone was.
  11. Materials and Methods: the authors do not appear to give information about a part of the instruments and drugs used. Please amend this point.

Minor comments:

  1. Line 29: not “was” but “were”.
  2. Line 31: “TNF-alpha” should be “TNF-a”.
  3. Line 139: please delete one of two “from”.
  4. Line 153: IbTx should be defined in this line but not in line 165.
  5. Line 179: please put “of” following “effect”.
  6. Line 200: “TNF-alpha” should be deleted.
  7. Line 205: not “figure 1a” but “figure 5a”.

Author Response

Dear and Reviewer,

The authors have carefully considered the stimulating criticism of the Referees. We have changed the manuscript at the points mentioned. Our revision includes all changes. We are convinced that the Reviewers’ suggestions have improved the quality of this manuscript.

Response to reviewers’ comments

Reviewer 2

Major comments:

  1. Line 111: there is no explanation about how 270 ± 2 pS was obtained. Please amend this point.

The conductance of the channel was calculated from the current-voltage relationship as before were described (Large-conductance Ca²⁺-activated potassium channel in mitochondria of endothelial EA.hy926 cells. Bednarczyk P, Koziel A, Jarmuszkiewicz W, Szewczyk A. Am J Physiol Heart Circ Physiol. 2013 Jun 1;304(11):H1415-27. doi: 10.1152/ajpheart.00976.2012.). We have added appropriate text to the Materials and Methods section and line 112.

  1. The last paragraph on page 4 and Fig. 2(b): statistical significance about dose-dependency for the naringenin effect is not clear. Please make this point clear.

To this point we have added statistical analysis (figure 2b). Also, we have added appropriate text to this paragraph.

  1. Line 147: ΔY should be defined here but not in line 353. Membrane but not membrane potential is depolarized. Line 151 and ΔY depolarization in the Y-axis of Fig. 3(a) should be revised.

The text and the figure has been corrected accordingly: “As shown in figure 3a, the addition of 5-50 μM naringenin significantly and dose-dependently decreased the mitochondrial ΔY by 0.65 ± 0.58 up to 7.01 ± 0.84 mV.”

  1. Line 153: it is not clear why paxilline at 0.3 mM was used. In line 124, paxilline (1 mM) is used. Alternatively, IbTx is used at 2 mM, a much larger value than that (0.3 mM) of paxilline. Please make these points clear.

We apologize for these errors. In single-channel recordings (line 124), 1 μM paxilline was used, while in experiments with isolated mitochondria 0.3 mM paxilline and 2 μM IbTx were applied. The text has been adjusted accordingly.

  1. Line 157 and 158: it is not clear from Fig. 3(b) whether a saturation effect is observed at 10 µM naringenin (the effect of naringenin at 10 µM is statistically larger in extent than that at 5 µM). Please make this point clear statistically.

The 10 µM naringenin concentration was the smallest concentration used resulting in the maximal effect on respiratory rate. However, there was indeed a significant difference between the change in respiratory rate after 5 µM naringenin and 10 µM naringenin P = 0.02. There were no significant differences between the 10 µM naringenin effect and the effects of higher flavonoid concentrations.

  1. The most left hand-side column of Fig. 4(a) does not appear to have SD. Please check whether this is right. The values of SDs of the results shown in Fig. 4 are very large. The last paragraph on page 6 should be revised by considering such large SD values.

The SD value in figure 4(a) has been corrected. When describing the results shown in Figure 4, we specifically emphasized, in line 186 at the end of page 6, that only results obtained under low calcium conditions are statistically significant.

  1. Content in lines 231-234 in Discussion section is quite the same as that in lines 70-72 in Introduction section. This point should be corrected.

Thank you. This point was rewritten.

  1. Lines 280-282: it would be better to mention a detail about the difference between vascular endothelial cells and skin fibroblasts (authors’ previous study) in mitoBKCa channel regulation by naringenin.

Thank you for the comment. These issue have been extended in the Discussion section.

  1. Do flavonoids other than narigenin activate mitoBKCa channels? Which structure of narigenin is involved in the activation of mitoBKCa channels? Please discuss this point.

Yes, we conduct new experiments with cardio-protective flavonoids and derivatives of the naringenin. This issue will be presented in the next manuscript. Also, we have started structural studies to extend issue according interaction of the flavonoids with mitochondrial potassium channels. In this year, very useful review according channel-flavonoids interaction wase published (The beneficial health effects of flavonoids on the cardiovascular system: Focus on K+ channels. Fusi F, Trezza A, Tramaglino M, Sgaragli G, Saponara S, Spiga O. Pharmacol Res. 2020 Feb;152:104625. doi: 10.1016/j.phrs.2019.104625.).

  1. Line 359: please mention the reason why rotenone was.

We studied Complex II driven respiration, using succinate as a respiratory substrate. Rotenone was used to inhibit Complex I of the electron transport chain. We have mentioned about it in Material and Methods.

  1. Materials and Methods: the authors do not appear to give information about a part of the instruments and drugs used. Please amend this point.

We have added an adequate information to the Materials and Methods section.

Minor comments:

  1. Line 29: not “was” but “were”.
  2. Line 31: “TNF-alpha” should be “TNF-a”.
  3. Line 139: please delete one of two “from”.
  4. Line 153: IbTx should be defined in this line but not in line 165.
  5. Line 179: please put “of” following “effect”.
  6. Line 200: “TNF-alpha” should be deleted.
  7. Line 205: not “figure 1a” but “figure 5a”.

All minor corrections have been made.

Attached manuscript contain changes according reviewer 1 and 2.

Round 2

Reviewer 2 Report

All of my concerns have been addressed and there is no concern in this revised manuscript except for several points that may serve to amend this version, as follows:

  1. Line 29: not “was” but “were”.
  2. Line 144; please put “of” following “presence”.
  3. Line 268: not “has” but “have”.
  4. Line 290: not “were” but “was”.